# Comparative Proteomic Analysis within the Developmental Stages of the Mushroom White *Hypsizygus marmoreus*

**DOI:** 10.3390/jof7121064

**Published:** 2021-12-11

**Authors:** Xiuqing Yang, Rongmei Lin, Kang Xu, Lizhong Guo, Hao Yu

**Affiliations:** 1Shandong Provincial Key Laboratory of Applied Mycology, School of Life Sciences, Qingdao Agricultural University, 700 Changcheng Road, Chengyang District, Qingdao 266109, China; yangxq@qau.edu.cn (X.Y.); 20192106292@stu.qau.edu.cn (K.X.); 198701007@qau.edu.cn (L.G.); 2Hubei Insect Resources Utilization and Sustainable Pest Management Key Laboratory, College of Plant Science & Technology, Huazhong Agricultural University, Shizishan Street, Wuhan 430070, China; rmlin@mail.hzau.edu.cn

**Keywords:** *Hypsizygus marmoreus*, proteomics, development, fruiting body, edible mushroom

## Abstract

(1) Background: The white *Hypsizygus marmoreus* is a popular edible mushroom in East Asia markets. Research on the systematic investigation of the protein expression changes in the cultivation process of this mushroom are few. (2) Methods: Label-free LC-MS/MS quantitative proteomics analysis technique was adopted to obtain the protein expression profiles of six groups of samples collected in different growth stages. A total of 3468 proteins were identified. The UpSetR plot analysis, Pearson correlation coefficient (PCC) analysis, and principal component (PC) analysis were performed to reveal the correlation among the six groups of samples. The differentially expressed proteins (DEPs) were organised by One-way ANOVA test and divided into four clusters. Gene Ontology (GO) and Kyoto Encyclopedia of Genes and Genomes (KEGG) analysis were performed to divide the DEPs into different metabolic processes and pathways in each cluster. (3) Results: The DEPs in cluster 1 are of the highest abundance in the mycelium and are mainly involved in protein biosynthesis, biosynthesis of cofactors, lipid metabolism, spliceosome, cell cycle regulation, and MAPK signaling pathway. The DEPs in cluster 2 are enriched in the stem and are mainly associated with protein biosynthesis, biosynthesis of cofactors, carbon, and energy metabolism. The DEPs in cluster 3 are highly expressed in the primordia and unmatured fruiting bodies and are related to amino acids metabolism, carbon and carbohydrate metabolism, protein biosynthesis and processing, biosynthesis of cofactors, cell cycle regulation, MAPK signaling pathway, ubiquitin-mediated proteolysis, and proteasome. The DEPs in cluster 4 are of the highest abundance in the cap and are mainly associated with spliceosome, endocytosis, nucleocytoplasmic transport, protein processing, oxidative phosphorylation, biosynthesis of cofactors, amino acids metabolism, and lipid metabolism. (4) Conclusions: This research reports the proteome analysis of different developmental stages during the cultivation of the commercially relevant edible fungi the white *H. marmoreus*. In the mycelium stage, most of the DEPs are associated with cell proliferation, signal response, and mycelium growth. In the primordia and unmatured fruiting bodies stage, the DEPs are mainly involved in biomass increase, cell proliferation, signal response, and differentiation. In the mature fruiting body stage, the DEPs in the stem are largely associated with cell elongation and increase in biomass, and most of the DEPs in the cap are mainly related to pileus expansion. Several carbohydrate-active enzymes, transcription factors, heat shock proteins, and some DEPs involved in MAPK and cAMP signaling pathways were determined. These proteins might play vital roles in metabolic processes and activities. This research can add value to the understanding of mechanisms concerning mushroom development during commercial production.

## 1. Introduction

*H. marmoreus* (Peck) H. E. Bigelow, also known as beech mushroom in some East Asian countries, is a white-rot fungus of the *Basidiomycota* [1]. It is popular for its unique seafood flavor, chewy texture, and nutritional characteristics [2]. So far, there are two commercial cultivated varieties, including the pale grey and white mushrooms [3]. *H. marmoreus* contains a variety of components with different biological activities, such as polysaccharides, (+)-catechin, gallic acid, and protocatechuic acid [4]. Polysaccharides and their derivatives own many functions. For example, polysaccharides from *H. marmoreus* have anti-inflammation and anti-oxidation effects [5]. A heteropolysaccharide, fucomannogalactan (FMG-Hm), exhibited promising anti-melanoma effects [6]. A non-lectin glycoprotein (HM-3A) was shown to be inhibitory against the growth of human myeloid leukemia U937 cells [7]. The water-unextractable proteoglycans exhibited immunomodulatory activities in vitro [8]. The angiotensin I-converting enzyme inhibitor displayed a clear antihypertensive action [9].

Worldwide, *H. marmoreus* is mainly obtained through industrial cultivation. Compared with other industrialized cultivation mushrooms, such as *Agaricus bisporus*, *Pleurotus eryngii*, and *Flammulina velutipes*, *H. marmoreus* requires more time (about 110 to 120 days) from inoculation to harvest. Therefore, the production cost of *H. marmoreus* is higher than other commercial mushrooms. It is essential to increase the growth rate of mushrooms and breeds. To this end, we should have a deep understanding of the life cycle of this mushroom and breed new cultivars accordingly. Until now, research on the developmental mechanisms of the *H. marmoreus* fruiting body have been very limited.

The fruiting body shape could be altered by changing carbon dioxide concentration in different growth stages [10]. The contents of free amino acids and soluble carbohydrates in the fruiting body of *H. marmoreus* also changed with morphogenesis [11]. Some proteins play vital roles in the physiological processes of *H. marmoreus*. For instance, carbohydrases such as xylanase, CM-cellulase, and amylase are important enzymes in the mycelial maturation and fruiting body growth of *H. marmoreus* [12]. Phosphoramidon is a specific metal proteinase inhibitor, which could completely inhibit the fruiting body formation. The addition of peptone and amino acids to a medium treated with phosphoramidon resulted in the increase in the dry weight of the fruiting body by 50% that of the control [13]. The *H. marmoreus* lcc1 gene was involved in mycelial growth and fruiting body initiation by increasing laccase activity [14].

With the rapid development of new technologies such as next-generation sequencing and MS-based omics techniques, a series of breakthroughs have been generated in “data-driven” biological research. Genomes of the main commercial mushrooms have been sequenced and annotated over the past few decades. Therefore, in recent years, omics studies such as comparative genomics, transcriptomics, proteomics, and metabolomics, have become indispensable methods to decipher the underlying mechanism of growth and development processes of mushrooms, to provide a theoretical basis for improving the cultivation process [15]. Fruiting body development, which requires the concerted action of several genes, is one of the most significant physiological processes for mushrooms. Several studies have been performed to analyze this differentiation process [15]. A reference strain of *H. marmoreus* (Haemi 51987-8) has been sequenced, assembled, and annotated, from which the carbohydrate-active enzyme has been analyzed to involve in the typical feature of white-rot fungi where auxiliary activity and carbohydrate-binding modules are enriched. Some trypsin genes and terpenoid biosynthesis gene clusters have also been discovered [16]. Recently, a chromosome level genome of *H. marmoreus* was assembled and annotated, from which a cytochrome P450 was identified as the candidate causal gene for the melanogenesis in this mushroom, based on bulked segregant analysis and comparative transcriptome analysis [17]. Interestingly, the aforementioned properties of *H. marmoreus*, which play regulatory roles in various stages of growth and development, can be attributed to its elusive metabolome and subtle transcriptome. The amino acid, nucleotide, and terpenoid metabolism-related metabolites genes are more abundant in the younger mushrooms [2]. Differentially expressed genes (DEGs) from the four transcriptomes of *H. marmoreus* elucidated the mechanisms of fruiting body development [18]. Zhang et al. suggested that light affects the expression of genes related to fruiting body initiation and nitrogen metabolism. The mTOR signaling pathway was also associated with promoting fruiting body maturation [18]. Generally, these studies provide meaningful information and a genetic basis for the research of the development of *H. marmoreus*.

Proteins are the executors of physiological functions. MS-based proteomics is a powerful tool for the direct characterization of the proteome of a biological sample, but cannot be carried out using genomic or transcriptomic techniques [19]. Proteomic techniques have been used for the study of fruiting body development, environmental factor response, secondary metabolites biosynthesis, and post-harvest [20,21,22]. In *F. velutipes*, 171 DEPs were identified from the samples of different development stages by the iTRAQ labeling technique. Several proteins, including mitogen-activated protein kinases (MAPKs) and heat-shock protein 70, were identified as stage-specific biomarkers to study the fruiting process for this mushroom [20]. Wang et al. identified 4380 proteins in the fruiting bodies of *Dictyophora indusiata* and concluded that cell wall stress-dependent MAPK pathway and cell wall degradation-related genes played vital roles in the morphological development of *D. indusiata* fruiting bodies [21]. Lectin, superoxide dismutase, and glycoside hydrolase are of great importance for the development of *Cordyceps militaris* [22]. Proteomic analysis is a powerful tool to understand the molecular mechanisms behind the fruiting body development of mushrooms. However, until recently, proteome analysis for the whole *H. marmoreus* fruiting body during its development stages has not been reported. In the present study, we adopted the Label-free LC-MS/MS quantitative proteomics analysis to detect the proteome of white *H. marmoreus* in the uninterrupted five stages of the fruiting body development process.

The determined DEPs and pathways could potentially play significant roles in the development of the fruiting body. Following this line of thought, the functions of many interesting proteins have been investigated. The purpose of this study is to contribute to the understanding of the proteomic profile of *H. marmoreus* during its developmental stages and provide information for further research on the growth and development mechanism of mushrooms.

## 2. Materials and Methods

### 2.1. Culture Conditions and Acquisition of the White H. marmoreus Samples

White *H. marmoreus* strain G12 was provided by Shandong Provincial Key Laboratory of Applied Mycology. The mycelia of this strain were maintained and cultivated as previously described [23]. The fruiting process was carried out in a mushroom factory in Dongying, Shandong Province, China. The spawn was inoculated in a culture medium and then cultivated at 25 °C for 80–90 days. After that, the surface mycelia were scratched and water was replenished into the cultivation bottle, and the date was set to be day 0. Mycelia were cultivated in a growth room with weak light, a temperature of 16~17 °C to 11~14 °C, carbon dioxide concentration of less than 1000 ppm, and humidity of 95~97%. The fungal samples were collected on day 13 (primordia group), day 14 (small unmatured fruiting body group), day 19 (larger unmatured fruiting body group), and day 23 (mature fruiting body groups) for the determination of fungal growth. There are four biological repeats for each stage from four mushroom growth bottles. The samples were snap-frozen in liquid nitrogen and stored at −80 °C until protein extraction.

### 2.2. Protein Extraction and Peptide Digestion

Total proteins were extracted from the frozen *H. marmoreus* samples according to the following protocol, 100 mg frozen sample was taken into the centrifuge tubes and then 1 mL UT buffer (8 M urea, 0.1 M Tris-HCl pH 8.5) containing HaltTM Protease inhibitor Cocktail (Thermo Fisher, Shanghai, China) was added. The tissueLyser II (QIAGEN, Hilden, Germany) was used to break the sample at 150 Hz for 60 s. The cell extract was treated by ultrasonication for 24 s (on for 6 s, off for 15 s). Tissue debris was removed by centrifugation (12,000× *g* for 10 min at 4 °C), and the supernatant was transferred into a new tube. Protein concentration was determined using the Micro BCA Protein Assay Kit (Thermo Fisher, Shanghai, China). After adding 15 mg DTT, the sample was incubated at 37 °C for 1 h.

Afterward, enzymolysis was performed according to the FASP method described by Wiśniewski et al. [24]. The extracted proteins (100 μg) dissolved in 300 μL UA buffer were taken into Pierce Protein Concentrators PES (10 K MWCO, 0.5 mL) (Thermo Fisher, Shanghai, China) to remove the low molecular impurities by centrifuging at 10,000× *g* for 30 min. 50 mM of iodoacetamide was added to alkylate the proteins for 30 min at room temperature in the dark. The proteins were washed with 200 μL UA and 300 μL of 50 mM NH_4_HCO_3_ after removing the buffer by centrifugation. 2 μg modified trypsin (Promega, Madison, WI, USA) in 100 μL of 50 mM NH_4_HCO_3_ was added into the ultrafiltration tube in a mass proportion of 1:50 (enzyme/protein). Enzymolysis was performed with gentle shaking at 37 °C for 12 h. After that, peptides were collected by centrifugation at 10,000× *g* for 15 min, and the residue peptides in the ultrafiltration tube were washed with 50 μL 50 mM NH_4_HCO_3_ for one more time. The salt of the pooled elutes was removed by using Merck Millipore ZipTip C18 resin (Darmstadt, Germany). Additionally, then, peptide concentration was measured by utilizing Pierce Quantitative Colorimetric Peptide Assay (Thermo Fisher, Shanghai, China). The peptide sample was lyophilized on an RVC 2-25 CD plus vacuum concentrator (Christ, Osterode am Harz, Germany), and subsequently stored at −80 °C for further analysis.

### 2.3. Label-Free LC-MS/MS Quantitative Proteomics Analysis

This analysis was performed with desalted peptides, which were reconstituted in 10 μL 0.1% formic acid, by using a Nano-LC system coupled with Orbitrap Fusion™ Tribrid™ (Thermo Fisher Scientific, San Jose, CA, USA). The MS instrument was operated in data-dependent acquisition mode, with full MS scans over a mass range of *m*/*z* 350–1500 with detection in the Orbitrap (120 K resolution) and with auto gain control set to 100,000. Different chromatographic gradient lengths from 60 to 240 min were tested for peptides separation. All gradient started at 5% (*v*/*v*) ACN (0.1% formic acid) and went up to 32% (*v*/*v*) ACN (0.1% formic acid). Three biological repeats for each group and two technical repeats for each sample were performed.

### 2.4. Peptides and Proteins Identification

The raw files were analyzed using Proteome Discoverer software suite version 2.0 (Thermo Fisher Scientific, San Jose, CA, USA) against the *H. marmoreus* proteome database (proteome ID UP000076154) from Uniprot. Protein identification was supported by at least two unique peptides with a false discovery rate lower than 0.05. The mass spectrometry proteomics data have been deposited to the *ProteomeXchange* Consortium via the PRIDE [25]. partner repository with the dataset identifier PXD028353.

### 2.5. Bioinformation Analysis

Raw data obtained from Proteome Discovery software was normalized as follows. Firstly, missing values were supplemented (four sets of data, data with only one set was deleted, missing values in data with two or three sets were supplemented, which is the k-proximity method). Then, median standardization was executed on intensity data. We used TBtools to construct heatmaps and enrichment analysis. UpSetR is an open-source R package that employs a scalable matrix-based visualization to show intersections of sets, their size, and other properties, which is available at https://github.com/hms-dbmi/UpSetR/ (accessed on 1 October 2021) [26]. The *H. marmoreus* proteome was annotated and functionally enriched using the GO tool (http://geneontology.org/) (accessed on 1 October 2021) according to cellular component (CC), molecular function (MF) and biological process (BP). We used Gene Set Enrichment Analysis (GSEA) for interpreting gene expression data [27]. We performed GO analysis by using the R-ggCyto tool [28]. Afterward, the KEGG pathways of candidate proteins were retrieved by blasting against the KEGG database (https://www.genome.jp/kegg/pathway.html) accessed on 1 October 2021) and mapped to pathways. DbCAN for carbohydrate-active enzyme (CAZyme) families (based on CAZyDB07/15/2016) [29] were used to annotate the identified proteins.

## 3. Results

### 3.1. Label-Free LC-MS/MS Quantitative Proteomics Analysis for the Six Groups of the White H. marmoreus Samples and the Correlation among the Six Groups of Samples

To gain insights into the events accompanying the transformation of mycelium into mature fruiting bodies in the white *H. marmoreus*. The growth process of the white *H. marmoreus* was divided into five stages according to its morphologicall changes: mycelium (Myc), primordia (Pri), small unmatured fruiting body (SUfb), larger unmatured fruiting body (LUfb), stem and cap of the mature fruiting body (Ste and Cap) (Figure 1). Protein expression profiles of the six groups of samples were obtained by adopting the Label-free LC-MS/MS quantitative proteomics analysis. A total of 3468 proteins were identified from all the six groups of samples, which were mapped to the total 16,572 peptide reads of *H. marmoreus* strain 51987-8 (Appendix A). The shared proteins were analyzed by UpSetR (Figure 2). Consequently, 1593 shared proteins were identified in all six groups of samples. The second largest subgroup containing 441 proteins was identified in all fruiting body (Pri, SUfb, LUfb, Cap, and Ste) groups.

PCC analysis (Figure 3a) indicated that the protein expression profile of the Myc group showed a higher correlation with that of the Ste group than other groups. Compared with the Ste group, the Myc group demonstrated a more divergent expression level. On the other hand, the protein expression profile of the Pri, SUfb, LUfb, and Cap groups displayed a higher correlation (>0.90) with each other. What is more, the PC analysis (Figure 3b) showed similar results. The results indicated that the main function of the stipe is elongation, similar to the function of the mycelia during vegetative growth. The primordia, the unmatured fruiting body, and the cap shared similar biological processes such as differentiation and pileus expansion. In addition, PCC analysis and PC analysis results revealed that the six groups of samples were reasonable with good correlations among biological repeats.

### 3.2. Six Groups of Samples Intersection and Quantitative Visualization of DEPs

To gain insight into the protein abundance change in different development stages or tissues, one-way ANOVA test controlled with BH FDR < 0.05 was performed to locate the DEPs (Appendix A). Before the test, the protein abundances was normalized by dividing the median value. Protein abundances with three missing values of one group were discarded. The *K*-nearest neighbor technique was used to estimate one or two missing values in one group.

Unsupervised hierarchical clustering further confirmed the correlation analysis and PC analysis results that the Myc group and Ste group were clustered, while the other four groups were clustered separately (Figure 4) (Appendix A). The heatmap reveals four main clusters of the DEPs. Cluster 1 contains the DEPs which have the highest abundance in the Myc group. Cluster 2 contains DEPs that are enriched in the Ste group. Cluster 3 represents the DEPs highly expressed in the primordia and unmatured fruiting bodies but not the Myc and Ste groups. Cluster 4 contains the DEPs that are only up regulated in the Cap group.

### 3.3. GO Analysis for the DEPs in Each Group to Molecular Function (MF), Cellular Component (CC), and Biological Process (BP) Categories

GO analysis was performed to classify the annotated DEPs for each cluster (Figure 5). These profiled DEGs were categorized into three main GO categories, CC, MF, and BP.

There are 674 DEPs in cluster 1 assigned to 29 GO terms (Figure 5a) (Appendix A). Ten of the GO terms were assigned to the BP category. Transmembrane transport (62, 9.2%), translation (25, 3.7%), intracellular protein transport (13, 1.9%), DNA-templated transcription (10, 1.4%), and carbohydrate metabolic process (9, 1.3%) were dominant terms. The CC category contains ten terms. Integral components of membrane (220, 32.6%), nucleus (35, 5.2%), endoplasmic reticulum membrane (19, 2.8%), and cytoplasm (15, 2.2%) were the most representative terms. There are nine terms in the MF category. ATP binding (48, 7.1%), DNA binding (29, 4.3%), metal ion binding (21, 3.1%), and transmembrane transporter activity (21, 3.1%) were the most common terms. These results suggest that the main DEPs in the Myc stage may be involved in cell proliferation and mycelium growth.

In cluster 2, there are 515 DEPs divided into 33 GO terms, including 12 BP, 12 CC, and 9 MF terms (Figure 5b) (Appendix A). Based on the number of DEPs classified into each category, the three largest categories were as follows: translation (28, 5.4%), transmembrane transport (13, 2.5%), and carbohydrate metabolic process (11, 2.1%) for BP; integral component of membrane (200, 38.8%), ribosome (41, 8.0%), nucleus (14, 2.7%) and cytoplasm (9, 1.7%) for CC; and ATP binding (36, 7.0%), structural constituent of ribosome (23, 4.5%), and metal ion binding (21, 4.1%) for MF. These DEPs mainly in the stem may be involved in the increase in biomass.

There are 1834 DEPs in cluster 3 grouped into 31 GO terms, comprising 11 BP, 10 CC, and 10 MF terms (Figure 5c) (Appendix A). The prominent terms of each category were as follows: translation (122, 6.7%), carbohydrate metabolic process (36, 2.0%), and protein folding (32, 1.7%) for BP; Integral component of membrane (147, 8.0%), nucleus (102, 5.6%), cytoplasm (102, 5.6%) and ribosome (54, 2.9%) for CC; and ATP binding (186, 10.1%), metal ion binding (89, 4.9%), oxidoreductase activity (51, 2.8%), RNA binding (48, 2.6%), and structural constituent of ribosome (46, 2.5%) for MF. These DEPs mainly in the Pri and Ufb stages may be involved in cell proliferation and cell proliferation.

In cluster 4, there are 426 DEPs divided into 29 terms, involving 12 BP, 7 CC, and 10 MF terms (Figure 5d) (Appendix A). The more conspicuous terms of each category were as follows: carbohydrate metabolic process (8, 1.9%), DNA-templated transcription (6, 1.4%), and protein transport (5, 1.2%) for BP; Integral component of membrane (83, 19.5%), nucleus (24, 5.6%), cytoplasm (13, 3.1%) for CC; and metal ion binding (21, 4.9%), ATP binding (18, 4.2%), zinc ion binding (17, 4.0%), RNA binding (10, 2.3%) for MF. These DEPs mainly in the cap may be involved in the biomass increase.

### 3.4. KEGG Pathway Enrichment Analysis of DEPs from Mycelium to the Mature Fruiting Body

To gain functional information about DEPs, KEGG pathway enrichment of DEPs was performed from multiple-sample tests (Figure 6). The results showed that the distributions of major enriched KEGG pathways in different clusters were different. The DEPs in cluster 1 were associated with 97 specific KEGG pathways, which were mainly ribosome, biosynthesis of cofactors, peroxisome, spliceosome, cell cycle and, MAPK signaling pathways (Figure 6a). The DEPs from cluster 2 were associated with 85 specific KEGG pathways, which were mainly distributed in the ribosome, biosynthesis of cofactors, carbon, and energy associated pathways (such as carbon metabolism, oxidative phosphorylation, TCA cycle, and starch and sucrose metabolism), and protein synthesis associated pathways (such as biosynthesis of amino acids, protein processing in the endoplasmic reticulum and protein export) (Figure 6b). A high abundance of DEPs enriched in these pathways satisfied the increase of biomass in the stem of the mature fruiting body. The cell cycle pathway was not the major pathway in cluster 2, which might indicate that the growth of the stem was mainly through cell elongation rather than cell division.

DEPs from cluster 3 were distributed in 109 KEGG pathways. Biosynthesis of amino acids and carbon metabolism pathways contains most DEPs, and other carbon and energy-associated pathways were also enriched in DEPs from cluster 3 (Figure 6c). The results are consistent with the fast growth of the fruiting body in these stages. In addition, ubiquitin-mediated proteolysis and proteasome are also the major enriched pathways in cluster 3. These might be associated with the stress response and differentiation of the fruiting body in these development stages. DEPs in cluster 4 were associated with 74 KEGG pathways. The major enriched pathways are spliceosome, endocytosis, nucleocytoplasmic transport, protein processing in the endoplasmic reticulum, oxidative phosphorylation, and biosynthesis of cofactors (Figure 6d). The involvement of spliceosome and endocytosis has been reported in the development of plants or animals, however, the role of these enzymes in mushroom development was not clear.

## 4. Discussion

### 4.1. The Proteome and Transcriptome of H. marmoreus Exhibit the Same Major Enriched Pathways and the Enriched Pathways Are Similar among Some Different Mushrooms

Although many transcriptomes and proteomes of fungi such as *P. ostreatus, Phanerochaete carnosa,* the hybridized *Trichosporon* fungi, *Lentinula edodes*, etc. have been reported [30,31,32], very few studies reveal the changes of expressed protein in the multiple successive stages of the cultivation process. Until this study, the protein profile changes of *H. marmoreus* from mycelium to the mature fruiting body were not reported. The whole genome and transcriptome of *H. marmoreus* has been assembled and annotated [17,18]. Transcriptome analysis showed that pathways related to “Translation”, “Protein-DNA complex”, “Transport”, “Ribosome biogenesis” and “Nucleosome” were significantly enriched during the transition from the primordium to the fruiting body in *H. marmoreus* [17]. The actively expressed KEGG pathways in the proteomes of *H. marmoreus* are similar to the pathways in the transcriptome. However, compared with its transcriptome or genome, there were fewer proteins found when mapping to the proteomes. The reasons for this need to be further studied. Since more and more macro-fungi proteomes are available [33], the enriched pathways and DEPs that are actively expressed could be compared among different fungi. In *F. velutipes*, proteins involved in carbohydrate metabolism, carotenoid formation, the TCA cycle, MAPK signaling pathway, the biosynthesis of fatty acids, and branched-chain amino acids are upregulated [20]. In *P. ostreatus*, key pathways in regulating the cap and stipe development, comprising starch and sucrose metabolism, sphingolipid metabolism, glycerophospholipid metabolism, and autophagy were enriched [34]. In the cultivation process of *H. marmoreus*, the pathways associated with amino acids metabolism, carbohydrate metabolism, lipid metabolism, MAPK signaling pathway, autophagy were enriched, which are similar to the pathways in *F. velutipes* and *P. ostreatus*. Generally, more knowledge of the proteins, such as proteins functions and the expression changes, might provide valuable information to research the molecular mechanisms of fruiting body initiation and development in basidiomycete fungi, especially industrially cultivated mushrooms.

### 4.2. MAPK and cAMP Signaling Pathways Play Vital Roles in the Growth and Development Process

The mushrooms were sensitive to environmental conditions. For instance, a light signal was necessary, otherwise, it would have developed into an undeveloped cap [32]. Low carbon dioxide concentration with high ventilation inhibits the differentiation toward the stipe and the cap [35]. The MAPK signaling pathway (Mkk1_2 and PBS2) and cAMP signaling (PKA) pathways might be associated with the growth and development of *H. marmoreus* [17].

MAPK pathways are important signal transduction pathways conserved in essentially all eukaryotes involved in the perception and adaptation of many species of fungus [36]. MAPKs are involved in regulating a wide variety of physiological activities, for instance, cell functions (proliferation, gene expression, differentiation, mitosis, cell survival, and apoptosis), cell wall integrity maintenance, fruiting body development, stress response, conidiation, etc. [21,37,38]. The DEGs involved in MAPK signaling pathways were identified in the white *H. marmoreus*. Three of them deserved our attention due to their high protein expression levels and significant differences among different groups. The first MAPK (A0A369K9Z4) was significantly highly expressed in the Myc group, which shows a high similarity with MAPK KSS1 (Saccharomyces cerevisiae S288C). The MAPK KSS1 acts in an antagonistic manner in response to mating factors to regulate the cell cycle [39]. This protein may be related to cell proliferation in the Myc stage. The second is MAPK (A0A369K4T8) with a higher expression in the Pri, SUfb, LUfb, and Cap groups, but lower in the Myc and Ste groups. Its closest analogy is MAPK HOG1 (*S. cerevisiae* S288C). This MAPK HOG1 enhances glycerol production by inducing the transcription of enzymes necessary for glycerol synthesis to increase glycerol production via directly regulating metabolism [40]. Glycerol is an important molecule for yeast metabolism and osmoadaptation [41]. It is speculated that cells increase intracellular osmotic pressure by increasing intracellular glycerol content to cope with various stresses, for example, drought, hypertonic environment, etc. The result reminds mushroom growers to pay more attention to water replenishment during the Myc and mature fruiting body stage. The third MAPK (A0A369JW23) was higher expressed in the Pri, SUfb, LUfb, and Ste groups. It is more similar to MAPK SLT2 (*S. cerevisiae* S288C). Signal transduction mediated by SLT2 is essential for maintaining cell wall integrity in *S. cerevisiae* [42]. The high expression of this MAPK may promote the active expression of enzymes related to the synthesis of cell wall components (chitin and dextran), such as the CAZYmes. Previous studies have suggested that CAZyme genes and MAPK can regulate the growth and development of edible fungi [43], which is consistent with our inferences.

The cyclic AMP (cAMP)/protein kinase A (PKA) pathway plays a key role in adaptation to nutrient availability. The impact of elevated cAMP is to activate cAMP-dependent protein kinase A (PKA), which then phosphorylates downstream target proteins. These proteins including enzymes, structural proteins, and transcription factors carry out a myriad of responses as a result of signaling through the pathway [36]. The Ras proteins are highly conserved GTPase signal transducers and transmit signals to downstream proteins by regulating cAMP synthesis. They are essential signaling regulators of fungi growth and development, and stress response in fungi [44,45]. Two Ras proteins (A0A369K612 and A0A369JVL6) were both higher expressed in the Ste group, indicating that the white *H. marmoreus* is more sensitive to some external pressure signals during the mature fruiting body stage. These signals may be crucial to the final mushroom morphology, especially the shape of stalks. A guanine nucleo-tide-binding protein subunit beta-like protein (GNBP: A0A369JUZ7) was highly expressed in the cultivation process, especially in the Pri, SUfb, LUfb, and mature fruiting body (Cap) stages. GNBP could increase cAMP level significantly and inhibit temperature-sensitive mutation of Ras proteins in *S. cerevisiae* [46]. In addition, several downstream proteins such as serine/threonine-protein kinase gad8 (A0A369K7I8 and A0A369J7Y6) and cAMP-dependent protein kinase regulatory subunit (PKRS: A0A369JT35) have a similar expression trend with GNBP. The gad8 is similar to serine/threonine protein kinase YPK1 (*S. cerevisiae* S288C), which can interact with rapamycin targets to regulate the salt stress response [47]. Ca^2+^ signaling (Ca^2+^-ATPase) might be more important in fruiting body maturation than in the other stages in *L. edodes* [32]. These results show that the white *H. marmoreus* is likely more susceptible to salt stress in the Pri, SUfb, LUfb, and mature fruiting body (Cap) stages. The PKRS is the most similar to cAMP-dependent protein kinase regulatory subunit BCY1 (*S. cerevisiae* S288C) [48]. cAMP-dependent protein kinase mediates many extracellular signals in eukaryotes. Bcy1 interacts with Ira2 tethering PICA to the Ras complex and Hsp60 chaperone localizes PICA to mitochondria and has a role in the kinase stability [49]. The glycogen accumulation and mitochondrial elongation phenotypes depend on Bcy1 [50]. These results show that multiple extracellular signals may be accepted mainly in the Pri, SUfb, LUfb, and mature fruiting body (Cap) stages.

Several interesting proteins are potentially related to fruiting body development, including carbohydrate-active enzymes (CAZymes), transcription factors (TFs), heat shock proteins (HSPs), and so on [51]. Functional analysis of these proteins may help reveal some novel aspects of known and important processes and provide new strategies for improving mushroom cultivation.

### 4.3. Identified CAZyme Families and Their Functions in the Growth and Development Process

CAZymes own two functions in a mushroom (*D. indusiate*), the cell wall remodels and degradation, and degradation of the culture substrate. Therefore, the CAZymes play significant roles in morphological changes and nutrient absorption [21]. Multiple CAZymes-related proteins were identified from the DEPs list by searching against the CAZyme database. Based on catalytic activities associated with conserved domains, they were classified as glycosyl hydrolases (GHs), glycosyl transferases (GTs), auxiliary activities (AAs), and so on.

There are two significantly higher expressed CAZymes in the Myc group, including cell surface mannoprotein MP65 (GH17: A0A369K3S0) and dolichol-phosphate-mannose-protein mannosyltransferase (GT39: A0A369JN93). O-mannosyltransferase belongs to a highly conserved protein family, which is responsible for the initiation of O-glycosylation of many proteins [52]. In the Myc stage, cell surface mannoproteins and glycosylated proteins were accumulated, which might provide a material basis for cell division and growth.

Two higher expressed CAZymes were identified in the Pri group, including 1,4-alpha-glucan-branching enzyme (GH13: A0A369JL46) and trehalose phosphorylase (GT4: A0A369JWT4). The 1,4-alpha-glucan-branching enzyme could introduce new branch points in starch by converting α-1,4-glucosidic linkages into α-1,6-glucosidic linkages, so it can produce modified starches [53]. Trehalose phosphorylase catalyzes the phosphorolysis of glycosides to produce saccharide 1-phosphate, promoting the catabolism of trehalose [54]. These results suggested that the primary degradation of carbohydrate compounds was more prominent in the Pri stage, preparing for the complete decomposition of these saccharide compounds to release energy.

There is a neutral trehalase (GH37: A0A369JUV5) screened in the SUfb group. Trehalases are highly conserved enzymes catalyzing the hydrolysis of trehalose in a wide variety of organisms. In the Pri and SUfb stages, trehalose-related metabolism was active, suggesting that planters could select the culture substrates rich in trehalose. The research indicates that the activity of yeast neutral trehalase is regulated in a calcium-dependent manner [55]. CaCl_2_ (EGTA) can affect stipe development by inhibiting the primordium differentiation in *P. ostreatus* [33]. Therefore, a proper amount of calcium might contribute to accelerating the growth and development of the white *H. marmoreus* by regulating the metabolism of trehalose.

In the LUfb group, seven CAZymes emerged as follows: beta-glucosidase (GH3: A0A369JFP3), endo glucanohydrolase (GH5: A0A369JWQ5), 1,4-alpha-glucan-branching enzyme (GH13: A0A369JL46), cell surface mannoprotein MP65 (GH17: A0A369JNY9), trehalose phosphorylase (GT4: A0A369JWT4), dolichol-phosphate-mannose--protein mannosyltransferase (GT39: A0A369JUK6), and mitochondrial cytochrome c peroxidase (AA2: A0A369J8H2). Beta-glucosidase and endo glucanohydrolase are important enzymes required for the hydrolysis of lignocellulosic biomass [56]. In the LUfb stage, glucosidase, hydrolase, glucan-branching enzyme, phosphorylase, mannosyltransferase are enzymes for the complete degradation of saccharides. One possible prediction was that as the fruiting body continues to grow, the more energy needed, the more saccharides degrade to meet energy needs.

In the Ste group, endo glucanohydrolase (GH5: A0A369JWQ5), alpha-amylase (GH13: A0A369K676), beta-1,3-glucan-binding protein (GH16: A0A369JKD7), chitin synthase (GT2: A0A369JK46), and pyranose dehydrogenase (AA3: A0A369J312) were screened out. Glucanohydrolase is a food enzyme cellulase [57]. The cellulase catalyzes the hydrolysis of 1,4-b-D-glucosidic linkages in cellulose, lichenin, and cereal b-D-glucans, resulting in the generation of mono, di-, tri-, tetra-, and oligosaccharides composed of glucose residues [58]. In this stage, the glucanohydrolase, binding protein, pyranose dehydrogenase might be utilized to degrade not only the cellulose in the culture medium for more energy but cell walls as well. The alpha-amylase might be used to degrade the starch for more energy. As we all know that the fungal cell wall structure is highly dynamic, changing constantly during its life cycle [21]. From the LUfb stage to the mature fruiting body stage, the fruiting bodies of the white *H. marmoreus* undergo great morphological changes, such as rapid stipe elongation. Chitin synthase might be utilized to synthesize one of the cell wall components—chitin.

The expression of Beta-glucosidase (GH3: A0A369JWE2), glycogen debranching enzyme (GH13: A0A369JN28) were prominent in the Cap group. Glycogen debranching enzyme (GDE), together with glycogen phosphorylase (GP), is responsible for the complete degradation of glycogen [59]. The primary-degraded carbohydrate metabolites might be degraded completely for enough energy.

The expression of CAZymes changes with the growth process of the white *H. marmoreus*, which implies that CAZymes play important roles in mushroom growth and development. This is consistent with previous studies [21,43,51].

### 4.4. Important TFs and Their Functions in the Growth and Development Process

TFs play vital roles in the development and the response of organisms to abiotic and biotic stresses [37]. 14 TFs of interest were identified from the DEPs list. There were two TFs (TF1: A0A369JP74 and TF2: A0A369JV45) play important roles in stress response. The TF1 is closest to Asg1p (*S. cerevisiae* S288C) NP_012136.1, which is involved in the transcriptional regulation of some stress response genes [60]. What’s interesting is that TF 1 was only detected in the Myc and LUfb groups. The TF2 is more similar to the kinase-regulated stress-responsive transcription factor SKN7 (*S. cerevisiae* S288C) NP_012076.3, which could regulate osmotic stress response genes and oxidative stress response genes [61]. However, TF2 is not detectable in the Pri and LUfb groups. The specific functions of both TFs remain to be further studied.

GATA TFs in fungi have various functions associated with light response, asexual-sexual development, etc [37]. A GATA TF (TF3: A0A369K1C3) was screened out in the Pri, Lufb and, mature fruiting body groups. This result indicates that the quality and growth cycle of the white *H. marmoreus* may be changed by adjusting the light during these three developmental stages.

Three heat shock factors (TF4: A0A369JTL2, TF5: A0A369JP53, TF6: A0A369JKY0) were found in the DEPs list. The most similar of them is stress-responsive transcription factor HSF1 (*S. cerevisiae* S288C) NP_011442.3, which can regulate transcription of heat shock genes and play a vital role in a wide variety of cellular processes including protein folding [62]. The trends of their expression are relatively consistent. They are more prominent in the Pri, SUfb, LUfb, and mature fruiting body (Cap) stages. Their effects on the growth and development of the white *H. marmoreus* may be explained by the expression and function of HSPs.

There are some novel TFs with obvious different expressions among the six groups, such as TF7: A0A369J7Y5, TF8: A0A369JGB7, TF9: A0A369KBU1, TF10: A0A369JIM1, TF11: A0A369JB62, TF12: A0A369J8P4, TF13: A0A369KD25, TF14: A0A369J7D1, which may function as regulatory components in the growth, development, and morphological changes of the white *H. marmoreus*. However, very little research on their functions could be found in the public databases. These proteins might be good candidates for in-depth experimental follow-up analyses to understand their specific roles.

### 4.5. HSPs and Their Functions in the Growth and Development Process

HSPs are highly conserved proteins in all organisms including microorganisms, plants, and animals, which play critical roles in regulating organism growth and development, as well as response to environmental stress [63]. Recent studies showed that heat shock proteins play important roles in mushroom development. In mice, HSP70 and HSP27 have been shown to play regulatory roles during erythropoiesis. The deactivation of the Hspd1 gene resulted in embryonic lethal [64]. The regulations of small heat shock proteins (sHsps) during development were reported in multiple organisms. A stage-restricted expression pattern of a subset of sHsps has been shown in different nematodes [65]. Studies about HSPs in the development of plants are just starting to emerge; still, several studies have revealed the involvement of HSP90s in the development of plants [66]. In the present study, several HSPs (A0A369JMQ9, A0A369JUF6, A0A369JNN0, A0A369JAL1, and A0A369K2K3) were upregulated in the primordia or young fruiting body samples compared with samples from the Myc group. The previous study by Liu et al. also detected the significant upregulation of HSP70 in the young fruiting body stage [33]. These results suggest that HSPs also play a role in mushroom development. The role of HSPs in response to stress has been proved in mushrooms. However, further studies are needed to analyze the function of HSPs in mushroom development.

## Figures and Tables

**Figure 1 jof-07-01064-f001:**
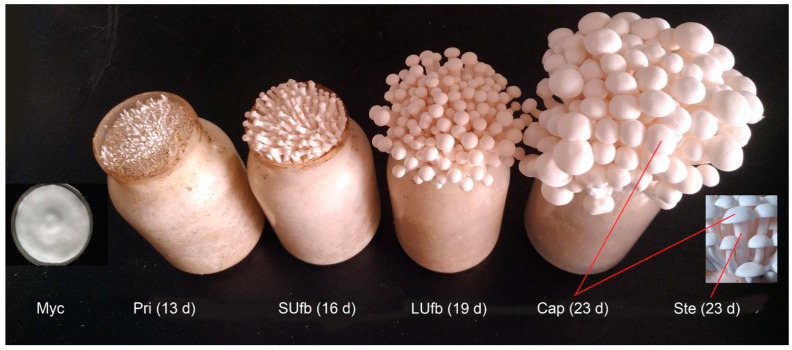
The growth process of the white *H. marmoreus* was divided into five stages after scratching the mycelium and six groups of samples were collected and detected for proteomic analysis. Myc: mycelium; Pri: primordia, on the 13th day after scratching; SUfb: small unmatured fruiting body, on the 16th day after scratching; LUfb: larger unmatured fruiting body, on the 19th day after scratching; Cap and Ste: cap and stem of mature fruiting body, on the 23rd day after scratching.

**Figure 2 jof-07-01064-f002:**
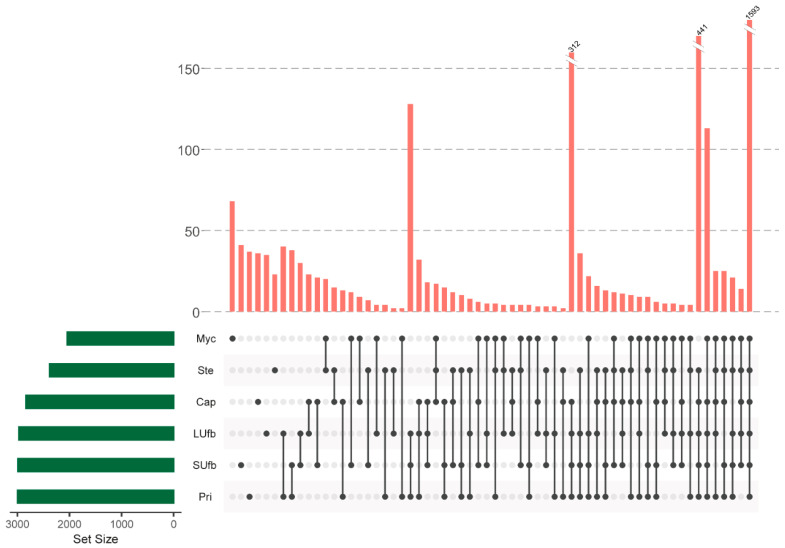
A UpSetR plot of variants across six groups of samples. The size of the intersections is shown as a bar chart placed on top of the matrix so that each column lines up with exactly one bar. A second bar chart showing the size of each group is shown to the left of the matrix. Each row represents one group, and the column represents their intersection in the matrix. A black-filled circle is placed in the corresponding matrix cell. A light grey circle shows that the group is not part of the intersection. A vertical black line connects the topmost black circle with the bottommost black circle in each column to emphasize the column-based relationships. Data for each column represents the number of expressed proteins that are contained in the groups corresponding to the black circle, which can be used to select proteins in particular intersections.

**Figure 3 jof-07-01064-f003:**
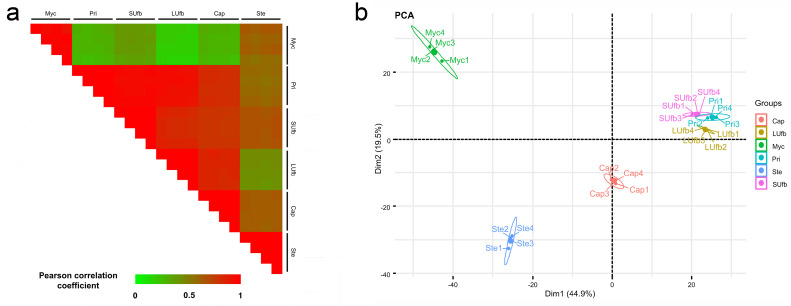
PCC analysis and PC analysis representation of the proteomic dataset of six groups of samples. (**a**) PCC analysis for pair-wise comparisons of proteome data. (**b**) PC analysis of proteome data from the six groups of samples. The Myc group and Ste group were located on one side of the first component zero-axis while the Pri, SUfb, LUfb, and Cap groups of samples were located on the other side.

**Figure 4 jof-07-01064-f004:**
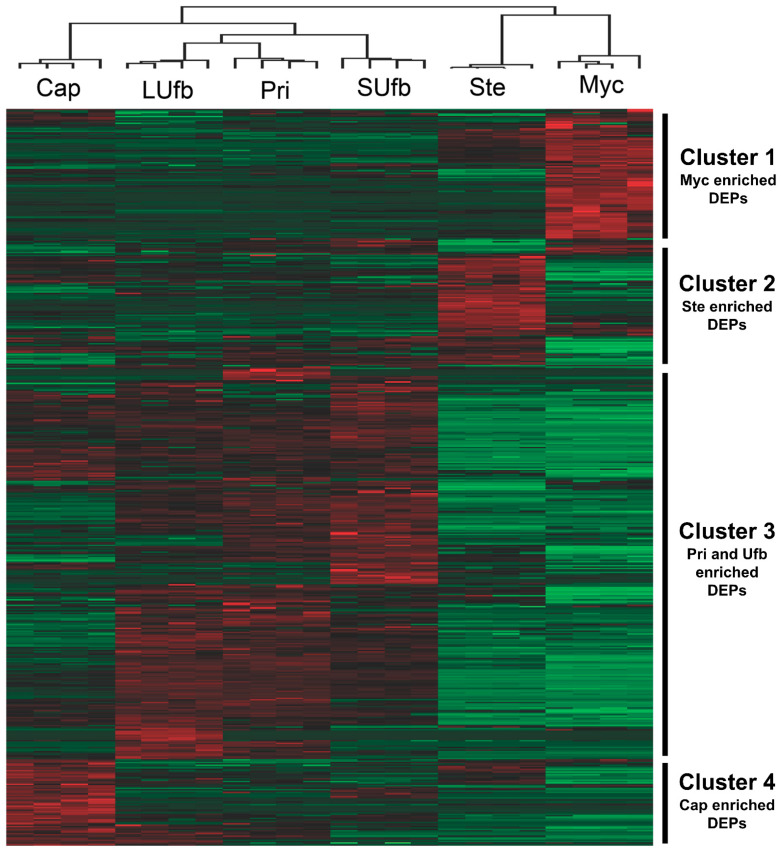
Unsupervised hierarchical clustering of DEPs by one-way ANOVA analysis (BH FDR < 0.05). The color code of each protein in all samples (columns) indicates the low (green) and high (red) Z-score normalized intensities.

**Figure 5 jof-07-01064-f005:**
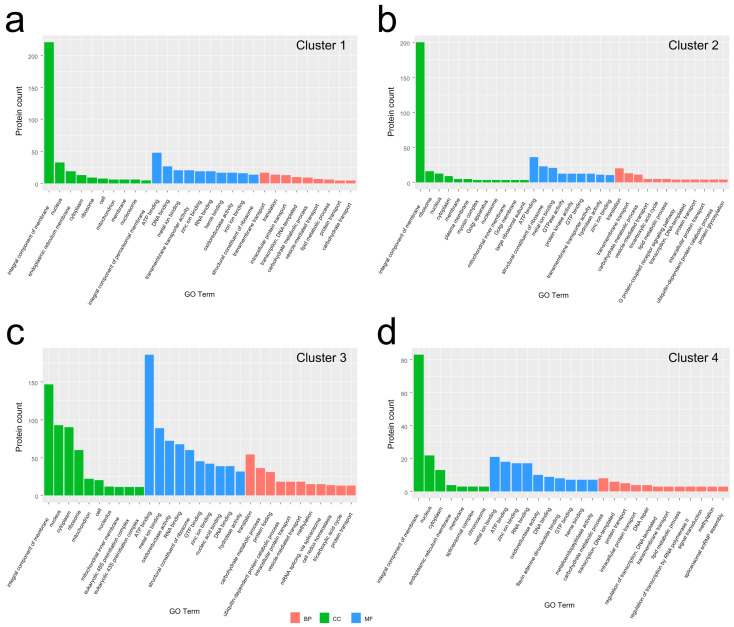
Enriched GO terms of DEPs from different clusters (**a**) Cluster 1, (**b**) Cluster 2, (**c**) Cluster 3, (**d**) Cluster 4. The cluster names were indicated in the bar plots.

**Figure 6 jof-07-01064-f006:**
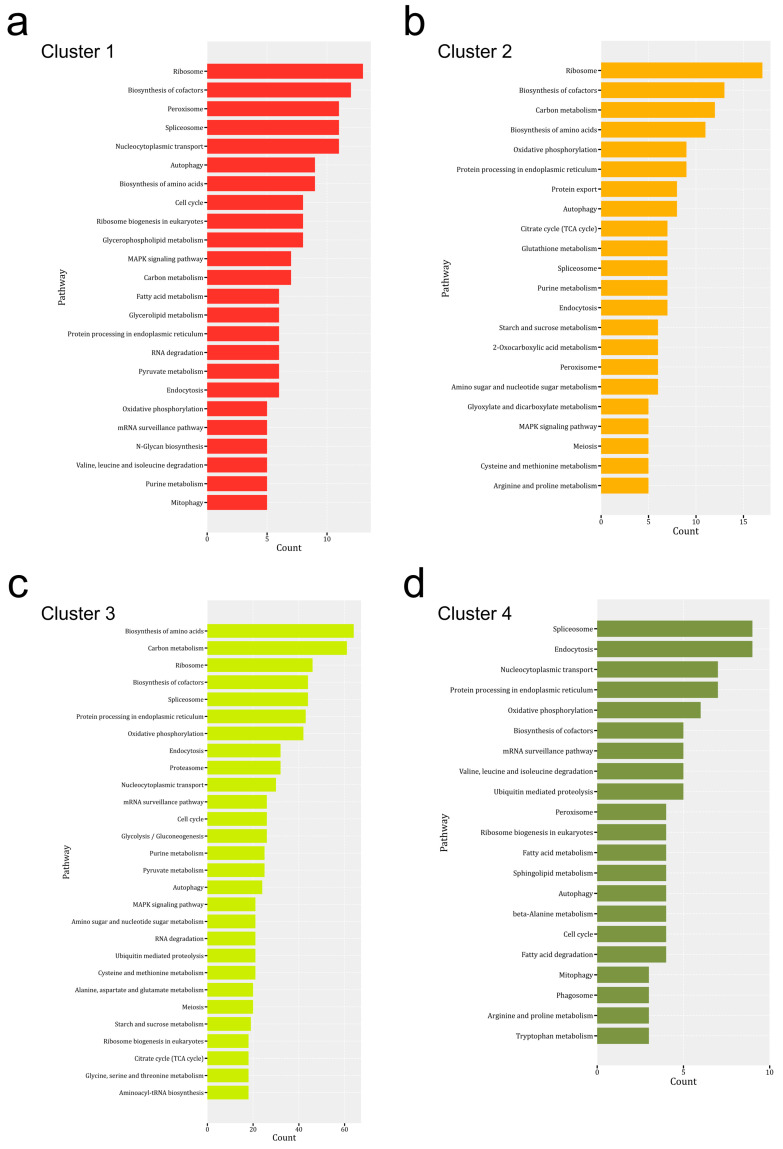
KEGG pathway enrichment of DEPs in four clusters ((**a**) Cluster 1, (**b**) Cluster 2, (**c**) Cluster 3, (**d**) Cluster 4) of Figure 3. The KEGG pathways with the most proteins were presented using a bar plot.

## Data Availability

Our raw data was named “Development of *Hypsizygus marmoreus*”, which has been successfully submitted to ProteomeXchange via the PRIDE database. The accession number is PXD028353.

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
