# Peer review of "Comparative Proteomic Analysis within the Developmental Stages of the Mushroom White Hypsizygus marmoreus"

_jof, 2021, doi:10.3390/jof7121064_

Round 1

Reviewer 1 Report

Review paper

General

The manuscript is interesting. The works are original and brings new information to the field. However, the writing and deliberation of the information were confusing, repetitive and not easy to understand, particularly with the use of uncommon terms and complex sentences. This also made the manuscript became lengthy and exhaustive. The reader would spend more time to understand the writing instead of the information. The author should improve the writing and the English in a more structured way. Please screen out unnecessary and non-relevant information. As an example, the authors could look into a paper by Amrah Weijn on melanin biosynthesis pathway in Agaricus bisporus mushrooms (Fungal Genet. Biol., 2013, 55-42-53).

The works had successfully indicated proteome profile during growth stages of the mushroom. However, the authors should discuss in more details about are intercorrelation of these proteins in the KEGG pathways. Especially, the authors have mentioned about obtaining more insights on the physiology of the mushroom growth for the benefit of improving the farming process.

From the list containing proteins and processes identified in the analysis (in figure 6), the authors could construct a hypothetical scheme for each developmental stage and compared that to that of other (edible) mushroom. The results contain many information that can be exploited; at present, the manuscript appears only to publish a data set.

Detailed comments.

Title

The title sounds odd because it conotates the dynamic of the analysis (instead of the proteome). Also, “cultivation process” refers to a treatment-based steps instead of natural. The title could be rephrase to “Analysis of proteome dynamic during the life-cycle of the mushroom Hypsizygus marmoreus”, “Proteome-dynamic analysis within the developmental stages of the mushroom Hypsizygus marmoreus”, or something alike.

Abstract

The author should be more specific. For example, onto which market is the mushroom common and popular, providing the trade/popular name is useful (e.g. champignon, shiitake, etc).

Elucidate developmental mechanism sounds not appropriate and not reconcile with the title; it should be to understand the proteomic profile during developmental stages or something like that.

Line 18. It should be “label free LC-MS/MS quantitative proteomic analysis”.

Line 19. The word continuous is redundant.

Line 18-23. Please rewrite because the information is scattered and sporadic. Group the information based on the subjects i.e. the sentence “the five developmental stages ... , respectively” is the subclause for the key stages in the previous sentence. The sentence “A total of 16,572 peptides reads ...” stands alone before the last sentence.

Line 34-35. What is KEGG enrichment? KEGG is a group of pathways, thus to enrich it, there should be an addition to the existing pathways or involvement of other pathway that overlaps with the pathways in KEGG. Taking from the rest of the sentence, the substituents/components of the pathways in KEGG were enriched (is this what the author mean?).

The conclusion should be specifically addressed to which mushroom (or is it for all mushroom in general?). Also, the current conclusion contains the aspiration, not conclusion of the work. It should mention how the proteome profile during the mushroom development stage, for example: in the early stage, most of the metabolites are associated with cell growth while those of in the later stage are towards expression of housekeeping proteins to facilitate the growth of mycelia, etc.

Introduction

line 44. Please describe the term “sister” in a more scientific way. Is it phenotypically, genealogy, physiologically, or morphology?

Line 47-48. What does this grey and white variant imply? Was the white derived from the grey one? Does the white still bear the same name or there would be notation e.g. “xxx sp” or “var xxx”? For example, Agaricus bisporus has several variant i.e. U1, burnetii, etc.

Line 48-56. Please write in a more direct and less defensive manner (i.e. repetitive use of the word proved and clear is unnecessary because it indicates lesser degree of confidence). Also, generally known/published information can be written in present tense except when it was aimed to describe the information in chronological order (which was not).

Line 57-60. The author did not mention the motivation to speed up the cultivation process (this can be called a process because it is part of farming; see comment to the title). Is it for economical reason?, is it to lower the risk of pathogenic attack that may damage the mushroom value?, or?

Line 67-68. It sounds odd: the major issue is having non-local strain for cultivation and production, but work to resolve the issue was not by exploration of local mushroom and their potential (instead of studying the proteomic of this selected strain from abroad).

Line 68-73. Genome sequencing and phenotyping is not technologies derived from edible fungi industry. Perhaps the author meant “these new technologies also brought opportunities to the edible fungi industry”; but then please explain how and why. What does the author mean with new way of thinking? Modifying the mushroom cultivation? Cultivation process improvement?

Line 74-75. Carbohydrates are not protein (and certainly not enzyme). Referring to the rest of the paragraphs, “some proteins” should be replaced with “biomolecules”.

Line 79. What does “fruiting uniform” mean?

Line 80-83. Please explain, what is the relevance of discussing phosphoramidon in this part? Is that implicating that inhibition or down regulation of certain enzyme/protein leads to a change in the growth rate? What is the hypothesis or known metabolic pathway with that incident? How is this information related to the next paragraph on signaling pathway during fruiting body formation?

Line 84-86. The author rightly mentioned that the genome and transcriptome information could be mined and analyzed, but this would only produce a frame work and hypothesis on development of the mushroom through its growth stages. This could not describe physiological traits, which should be assessed through experiments. Please carefully define, which information is eligible for what.

Line 89-91. The paragraph started with an indication that the genome and transcriptome are established. Why here is stated that the progress was preliminary?

Line 92-93. Please describe what the typical features are, why it is typical of white-rot fungi? Please describe auxilliary activity, what does it mean with enrichment: more genes, upregulated, or? Also, enriched in comparison to what?

Line 95. The word subsumed is of unusual use (please use common expression), the word elusive and subtle are not necessary.

Line 98. Small size mushroom is referring to the size of H. marmoreus fruiting body or other mushroom; in case of the latter, how is that relevant?

Line 98-102. Is this published information or a leak to the results from the present study? The two sentences appeared not correlate.

Line 107. The sentence is wordy and confusing; better to write “... analysis for the whole H. marmoreus fruiting body during its development stages ....” .

Line 108-109. The sentence does not make sense. The proteome might help who? If it were for “us” , what does achieve accurate regulation refer to? The government regulatory body to control the mushroom farming industry?

Line 112. If there were 5 key stages, how many stages actually are and why these 5 are the keys (among them)?

Line 112-115. “By comparing ... these stages.” Who is the subject of this sentence?

Material and methods

Line 131. What does four biological parallels mean? Four samples of different stages? Done in quadriplicate (to obtain statistics)? Four different mycelia preparation?

Line 134. What is UA abbreviated from?

Line 191-192. What are GO and KO abbreviated from

Results

Line 198. Please write the subsection title.

Line 201. Please mention the 80-90 unit (hour, day, week, or?)

Line 202. What is the definition of “fast”, in comparison to what and what is the standard to categorize it as “fast”. Why did not immediately mention 23 days (which according to the introduction is actually considered “long”).

Line 202-217. This paragraph is more like an extended part of methods (it should be incorporated there). There is no observation or measurement. The authors could instead write the observation during these growth stages, for example, was there browning, what is the average size of the sample (since the size has something to do with the metabolite content, as mentioned in the introduction).

Line 221-225. This part is unnecessary, it has been repeatedly mentioned in the previous sections.

Line 227. The number 51987-8 refers to what, strain number/code/variant?

Line 228-229. This is already shown by the statistics (p<0.05), it should be expressed with that.

Line 232. Replace “instead of” with “than” because it is a comparison, not a selection or preference.

Line 243. What does “significantly differentially expressed proteins” mean? Is it different proteins with significant expression, proteins with significantly different expression, or? Does the author mean significantly DEP? Then, what was the significantly term referring to? Differentially means the spectral count ratio is more than 2, thus the two protein samples are different. Does non-significantly DEP term exist?

Line 244. What is normalized median methods? This sentence is not clear.

Line 247-254. Most of the text should become the figure legend because it explains how to read the graph.

Line 293-294. Please use the term DEP consistently. Once it is abbreviated, use the abbreviation throughout the rest of the manuscript (otherwise it will be regarded as something else). Change the word “till” with “to”.

Line 302. Please expand the abbreviated “GO” because the abbreviation is only mentioned afterwards. Abbreviation should mentioned first before consistently used. It has already been abbreviated in material and methods (but no explanation on what it is abbreviated from).

Line 304-305. Please mention what is included in “so on”, otherwise just end with peptide sequences and genes if those in the “so on” are not relevant. The sentence “gene ontology .... terms to gene” does not make any sense.

Line 308-309. Please explain how the author could enriched the proteins? Was the enrichment not observed from the natural growth of the mushroom? Was there any treatment (e.g. heating, exposure to light, high salt, whatever) to increase the expression of certain proteins thus they would be found to be enriched during the analysis? In case of the latter, please add the treatment in the methods section.

Line 318-319. What/who are A0A369K2I0, ..., A0A369JNI4? Gene annotation? Protein code? If they were processes, what processes are they? They are not included in figure 6A.

Line 330-331. See the comment for line 318-319.

Line 332. Please mention what the materials are.

Line 343-344. See the comment for line 318-319.

Line 345-346. Please mention what the materials are. How the did author come into conclusion that the cells are going into Sufb stage, is there any correlation to that stage? Does the last sentence imply that efforts in the other stages are not laying good foundation for the mushroom growth?

Line 356-357. See the comment for line 318-319.

Line 358. See the comment for line 345-346 but for Lufb stage.

Line 373. See the comment for line 318-319.

Line 374-375. How did the author come into a conclusion that a lot of proteins stored and please describe what are “life activities”.

Line 388-389. See the comment for line 318-319.

Line 389-390. How did the author come into this conclusion?

Line 391-394. The information in line 310-390 could simply be presented in one table.

Line 395-396. Why this section has the same subtitle as 3.1.4?

Line 398-399. Phe, Tyr and Trp are amino acids, what is the different to the amino acid mentioned later on in this sentence? Please explain what does biosynthesis of ribosome mean.

Line 401. Please indicate, onto which subject the word “respectively” is referring to.

Line 403-410. What does Cap-up and Ste-up mean?

Line 403-414 and Figure 7. The text is not helpful to decribe the figure and each point in the figure has no annotation (what is what). The sequence of kinesin motor, catalytic domain, etc is not of equal value/group. Catalytic domain is a functional part of a protein, cytoskeleton-associated proteins is a grouping based on activity, while WD40 and HAT repeats are of protein structural motifs. Also, please indicate, onto which subject the word “respectively” is referring to.

Discussion

Line 453. Please mention which fungi thus the readers do not have to read the reference to find it.

Line 461. Please describe what and which are the KEGG pathways found during this proteome study and their comparison to the transcriptome study. Was all components recovered?

Line 463-464. Please give a support for this statement (a study in other fungi or microbe). Why limitation in the number of defined/elucidated protein was not considered as one of the causes?

Line 468-473. Please rewrite this part because it is confusing. The author could write: “In F. Velutipes (-common name-), protein involved in carbohydrate metabolism, carotenoid formation ..... and brached chain amino acids are up regulated [31]”.

Line Line 476-479. These pathways are not explicitly shown in the result section and immediately compared to the pathways in F. velutipes and P. ostreatus. There should be adjustment between them; for example carbohydrate metabolism is clearly mentioned (line 471, 474, and 477) but not much about e.g. autophagy (equal to endocytosis?) or spliceosome.

Line 480. Which knowledge of proteins was the author refering to?

Line 481-482. To come into this closure, the author should describe similarity in developmental stages of the fruiting bodies for various basidiomycetes. Only then the comparison on proteome profile could be correlated and compared in par (between spore and mycelia formation, fruiting body maturation, etc.).

Line 485-486. CaCl2 and EGTA are two different compound with different effects. Was this in combination or each of them?

Line 487. What does high ventilation mean?

Line 494-495. How the authors could enriched the signaling proteins? Which part in the methods was aimed for this treatment? The proteome data suggests that the signaling proteins were elevated, not enriched.

Line 495-499. Expression of these proteins were claimed to be increased. How was their expression when the mushroom is grown under standard conditions, thus no excess of external interference like high salt and/or exposure to light?

Reviewer 2 Report

Dear authors,

This manuscript reports the proteome analysis of different developmental stages during the cultivation of the commercially relevant edible fungi white Hypsizygus marmoreus (white beech mushroom or white shimeji). The article is poorly written an requires extensive revision of the language to favour the text flow. Besides the authors do not explain the characteristics of the substrate employed for cultivation, something that, when talking about heterotrophic organisms can v¡be relevant to understand the research conducted.

Besides the discussion is very poor and the article is mostly focused on describing the findings through the six differential developmental stages evaluated other than comparing results to the state-of-the-art which can very much maximise the impact of the study. For instance relevant literature on the topic refers to the widely studied button mushroom (Agaricus bisporus): Baars et al. (2020). Molecules, 25(13), 2984; Eastwood et al., 2013. Fungal Genetics and Biology, 55, 54-66. In summary the discussion only includes ref. 26-33 which appears very poor for this reviewer concerning the significant amount of data generated.

Overall this article must be rejected in the present form and the authors must be encouraged to try resubmission after major revision, since the content is relevant and can add value for the understanding of mechanistics concerning muhsroom development during commercial production.

Figure 3 is different to understand maybe further explanation requires in the text. Figure 4 requires that the acronyms of the developmental stages are summarised. Probably the figure will benefit of compiling Myc vs other … and so on in separate rows, depending on the ref. factor. I do not understand the relevance of Figure 5. Why are you only including cap and Ste? Figure 6 “GO enrichment analysis statistics of…”, this is redundant since that explanation is in the first line of the figure legend. Figure 7 is very little informative. Explanation required.

Besides some further comments:

Line 16- Need to be rephrase: “Until recently,…”

Line 22- This is not required information for the abstract “Data of proteomes…”

Line 43-45- Rephrase.

Line 46, 47 – “grey”

Line 50- were

Line 51- Rephrase

Line 57- “Rephrase “has been becoming”

Line 62.

Line 79 – fruiting uniform

Line 88- “to involve”

Line 131- Replicates

Line 144- Wisniewski et al. (2009)

Line 199- Section 3.1.1 is explaining M&M more than results, and it is not relevant, this must be eliminated.

Line 234 – “inferior to it” - Rephrase

Section 3.1.3- Huge explanation to understand Fig. 3.

Line 392- The authors reflect upregulated and downregulated genes in a difficult way, are you meaning differentially expressed genes?

Line 344- significant enrichment.

Section of discussion- I am not entirely sure to understand whether the tittles of different subsections efficiently summarised the content of the paragraphs. This must be reviewed.

Author Response

Response to Reviewer 2 Comments

Dear authors,

Point 1) This manuscript reports the proteome analysis of different developmental stages during the cultivation of the commercially relevant edible fungi white Hypsizygus marmoreus (white beech mushroom or white shimeji). The article is poorly written an requires extensive revision of the language to favour the text flow. Besides the authors do not explain the characteristics of the substrate employed for cultivation, something that, when talking about heterotrophic organisms can v¡be relevant to understand the research conducted.

Response: Thank the reviewer for the comment. We have made extensive revisions to the article, including the analysis results and language.

Possibly considering the pressure of commercial competition, the mushroom factory did not provide a specific medium formula.

Point 2) Besides the discussion is very poor and the article is mostly focused on describing the findings through the six differential developmental stages evaluated other than comparing results to the state-of-the-art which can very much maximise the impact of the study. For instance relevant literature on the topic refers to the widely studied button mushroom (Agaricus bisporus): Baars et al. (2020). Molecules, 25(13), 2984; Eastwood et al., 2013. Fungal Genetics and Biology, 55, 54-66. In summary the discussion only includes ref. 26-33 which appears very poor for this reviewer concerning the significant amount of data generated.

Response: Your suggestion is greatly appreciated. We have rephrased and supplemented the discussion section. Five subsections were set and listed below.

4.1. The proteome and transcriptome of H. marmoreus exhibit the same major enriched pathways, the pathways are similar among some different mushrooms

4.2. MAPK and cAMP signaling pathways play vital roles in the growth and development process

4.3. Identified CAZyme families and their functions in the growth and development process

4.4. Important TFs and their functions in the growth and development process

4.5. HSPs and their functions in the growth and development process

Point 3) Overall this article must be rejected in the present form and the authors must be encouraged to try resubmission after major revision, since the content is relevant and can add value for the understanding of mechanistics concerning muhsroom development during commercial production.

Response: Thank the reviewer, your opinions inspired us and we revised the manuscript accordingly. We have made a major revision to the article, please continue to comment on the revised manuscript.

Point 4) Figure 3 is different to understand maybe further explanation requires in the text. Figure 4 requires that the acronyms of the developmental stages are summarised. Probably the figure will benefit of compiling Myc vs other … and so on in separate rows, depending on the ref. factor. I do not understand the relevance of Figure 5. Why are you only including cap and Ste? Figure 6 “GO enrichment analysis statistics of…”, this is redundant since that explanation is in the first line of the figure legend. Figure 7 is very little informative. Explanation required.

Response: Thank you for the comment. We recombed the methods and results.

Methods: Label-free LC-MS/MS quantitative proteomics analysis technique was adopted to obtain the protein expression profiles of the six groups of samples collected in different growth stages. A total of 3,468 proteins were identified. The UpSetR plot analysis, Pearson correlation coefficient (PCC) analysis, and principal component (PC) analysis were performed to reveal the correlation among the six groups of samples. The differentially expressed proteins (DEPs) were sorted out by One-way ANOVA test and divided into four clusters. Gene Ontology (GO) and Kyoto Encyclopedia of Genes and Genomes (KEGG) analysis were performed to divide the DEPs into different metabolic processes and pathways in each cluster.

Results: The DEPs in cluster 1 are of the highest abundance in the mycelium and are mainly involved in protein biosynthesis, biosynthesis of cofactors, lipid metabolism, spliceosome, cell cycle regulation, and MAPK signaling pathway. The DEPs in cluster 2 are enriched in the stem and are mainly associated with protein biosynthesis, biosynthesis of cofactors, carbon and energy metabolism. The DEPs in cluster 3 are highly expressed in the primordia and unmatured fruiting bodies and are related to amino acids metabolism, carbon and carbohydrate metabolism, protein biosynthesis and processing, biosynthesis of cofactors, cell cycle regulation, MAPK signaling pathway, ubiquitin-mediated proteolysis, and proteasome. The DEPs in cluster 4 are of the highest abundance in the cap and are mainly associated with spliceosome, endocytosis, nucleocytoplasmic transport, protein processing, oxidative phosphorylation, biosynthesis of cofactors, amino acids metabolism, and lipid metabolism.

Besides some further comments:

Point 5) Line 16- Need to be rephrase: “Until recently,…”

Response: The reviewer’s suggestions have been adopted.  “Until recently, …”

has been changed to

“…. market. Researches on the systematic investigation of the protein expression changes in the cultivation process of this mushroom are few.”.

Point 6) Line 22- This is not required information for the abstract “Data of proteomes…”

Response: The reviewer’s suggestion has been adopted. “Data of proteomes…” was deleted.

Point 7) Line 43-45- Rephrase.

Response: Thank you for reminding us of the improper description.

This part has been rephrased into

H. marmoreus (Peck) H. E. Bigelow, also known as beech mushroom in some East Asian countries, is a white-rot fungus of the Basidiomycota [1]. It is popular for its unique seafood flavor, chewy texture, and nutritional characteristics [2].”.

Point 8)Line 46, 47 – “grey”

Response: Thank you for reminding us of the improper description. The “gray” has been changed to “grey”.

Point 9) Line 50- were

Response: Thank you for reminding us of the improper description.

“Previous studies have reported that H. marmoreus contained a variety of components, such as (+)-catechin, gallic acid, and protocatechuic acid, which were were proved to have different biological activities [4].”

has been rephrased into

H. marmoreus contains a variety of components with different biological activities, such as polysaccharides, (+)-catechin, gallic acid, and protocatechuic acid [4].”

Point 10) Line 51- Rephrase

Response: Thank you for reminding us of the improper description.

“Polysaccharides from H. marmoreus were proved to own the characterization of anti-inflammation and anti-oxidation [5]. A heteropolysaccharide, fucomannogalactan (FMG-Hm), exhibited promising anti-melanoma effects [6].”

has been rephrased into

“Polysaccharides and their derivatives own many functions. For example, polysaccharides from H. marmoreus have anti-inflammation and anti-oxidation effects [5]. A heteropolysaccharide, fucomannogalactan (FMG-Hm), exhibited promising anti-melanoma effects [6].”

Point 11) Line 57- “Rephrase “has been becoming”

Response: Thank you for reminding us of the improper description.

“Nowadays, the H. marmoreus mushroom has been becoming a more and more popular food around the world.” has been deleted.

Point 12) Line 62.

Response: Thank you for reminding us of the improper description. This part has been rephrased as

“To this end, we should have a deep understanding of the life cycle of this mushroom and breed new cultivars accordingly. Until now, researches on the developmental mechanisms of the H. marmoreus fruiting body are very limited.”

Point 13) Line 79 – fruiting uniform

Response: Thank you for reminding us of the improper description. The “fruiting uniform” has been changed to “fruiting body shape”.

This part has been rephrased into

“The fruiting body shape could be altered by changing carbon dioxide concentration in different growth stages [10].”.

Point 14) Line 88- “to involve”

Response: Thank you for reminding us of the improper description.

“The H. marmoreus lcc1 gene was proved to involve in mycelial growth … [14].”

has been rephrased into

“The H. marmoreus lcc1 gene was involved in mycelial growth … [14].”.

Point 15) Line 131- Replicates

Response: Thank you for reminding us of the improper description. The word “replicates” has been changed to “repeats” in the whole text.

Point 16) Line 144- Wisniewski et al. (2009)

Response: Thank you for reminding us of the improper description. “Jacek R WiÅ›niewski [19].” has been changed to “WiÅ›niewski et al. [24].”. More citations were added and the serial numbers were rearranged.

Point 17) Line 199- Section 3.1.1 is explaining M&M more than results, and it is not relevant, this must be eliminated.

Response: Thank you very much for the reviewer’s good suggestions. This section has been eliminated. And some information was transferred to “2. Materials and Methods”.

Point 18) Line 234 – “inferior to it” – Rephrase

Response: Thank the reviewer for the comment and we revised the manuscript accordingly. The “inferior to it” has been deleted.

This paragraph was rephrased as

“PCC analysis (Figure 3a) indicated that the protein expression profile of the Myc group showed a higher correlation with that of the Ste group than other groups. Compared with the Ste group, the Myc group demonstrated a more divergent expression level. On the other hand, the protein expression profile of the Pri, SUfb, LUfb, and Cap groups displayed a higher correlation (> 0.90) with each other. What is more, the PC analysis (Figure 3b) showed similar results. The results indicated that the main function of the stipe is elongation similar to the function of the mycelia during vegetative growth. The primordia, the unmatured fruiting body, and the cap shared similar biological processes such as differentiation and pileus expansion. In addition, PCC analysis and PC analysis results revealed that the six groups of samples were reasonable with good correlations among biological repeats.”.

Point 19) Section 3.1.3- Huge explanation to understand Fig. 3.

Response: Thank the reviewer for the comment and we revised the manuscript accordingly. 

This section was rephrased as

“Protein expression profiles of the six groups of samples were obtained by adopting the Label-free LC-MS/MS quantitative proteomics analysis. A total of 3,468 proteins were identified from all the six groups of samples, which were mapped to the total 16,572 peptide reads of H. marmoreus strain 51987-8 (Supplemental Table S1). The shared proteins were analyzed by UpSetR (Figure 2). Consequently, 1593 shared proteins were identified in all six groups of samples. The second largest subgroup containing 441 proteins was identified in all fruiting body (Pri, SUfb, LUfb, Cap, and Ste) groups.”.

The following part has been transferred to the legend of Figure 2.

“Each row represents one sample and the column represents their intersection in the matrix. A black-filled circle is placed in the corresponding matrix cell. A light grey circle shows that the sample is not part of the intersection. A vertical black line connects the topmost black circle with the bottommost black circle in each column to emphasize the column-based relationships. Data for each column represents the number of expressed proteins that are contained in the samples corresponding to the black circle, which can be used to select proteins in particular intersections.”.

Point 20) Line 392- The authors reflect upregulated and downregulated genes in a difficult way, are you meaning differentially expressed genes?

Response: Thank you for reminding us of the improper description. The expression of the original sentence is ambiguous, so it was deleted. A new GO analysis was performed (Figure 5) and this part was rephrased in the revised manuscript. Several groups of DEPs could be screened out and grouped into three categories.

Point 21) Line 344- significant enrichment.

Response: Thank you for reminding us of the improper description. This part was rephrased and transferred to the “4. Discussion”.  

Point 22) Section of discussion- I am not entirely sure to understand whether the tittles of different subsections efficiently summarised the content of the paragraphs. This must be reviewed.

Response: Thank the reviewer for the comment. The discussion has been revised accordingly. And the tittles of different subsections could summary the content of the paragraphs efficiently. Please comment on the revised manuscript.

Reviewer 3 Report

The paper mainly describes the proteomics analyses of Hypsizygus marmoreus in the fruiting body development process, to elucidate the development mechanism. the experiments themselves are quite typical and the analytic data are reliable. However, I would like to raise several concerns as follows about the research and thus if the quality of the paper is heightened. My comments are shown below.

1) Which specific proteins (enzymes) and metabolic pathways were found to play important roles in which stages? The author should carefully and in detail describe the relationship between the unique proteins and the morphological changes during the growth process of fruiting bodies (e.g., development of fruiting body, elongation of the stem, differentiation and swelling of cap, etc.).

2) L75 “carbohydrates .....,”

It should be revised to “carbohydrates active enzymes such as xylanase, CM-cellulase and amylase,”.

3) L147 “30min”

It should be revised to “30 min”.

4) L149, L150 “NH4HCO3”

It should be revised to “NH4HCO3”.

5) L418

What do you mean by "the 0th day of fungal growth"?

Author Response

Response to Reviewer 3 Comments

Review 3

The paper mainly describes the proteomics analyses of Hypsizygus marmoreus in the fruiting body development process, to elucidate the development mechanism. the experiments themselves are quite typical and the analytic data are reliable. However, I would like to raise several concerns as follows about the research and thus if the quality of the paper is heightened. My comments are shown below.

Point 1)  Which specific proteins (enzymes) and metabolic pathways were found to play important roles in which stages? The author should carefully and in detail describe the relationship between the unique proteins and the morphological changes during the growth process of fruiting bodies (e.g., development of fruiting body, elongation of the stem, differentiation and swelling of cap, etc.).

Response: We thank the Reviewer for the positive comments. The discussion part has been rewritten accordingly.

Point 2)  L75 “carbohydrates .....,”   It should be revised to “carbohydrates active enzymes such as xylanase, CM-cellulase and amylase,”.

Response: Reviewer’s suggestions have been adopted. “carbohydrates .....,” has been changed to “carbohydrases such as ....”.

Point 3)  L147 “30min”   It should be revised to “30 min”.

Response: Reviewer’s suggestions have been adopted. “30min” has been changed to “30 min”.

Point 4)  L149, L150 “NH4HCO3”  It should be revised to “NH4HCO3”.

Response: Reviewer’s suggestions have been adopted. “NH4HCO3” has been changed to “NH4HCO3”.

Point 5) L418  What do you mean by "the 0th day of fungal growth"?

Response: Thanks for the comment, and the “the 0th day of fungal growth” has been deleted from the revised manuscript.

Round 2

Reviewer 2 Report

Dear Editors,

This is a second round of revision from the Manuscript ID: jof-1441447.

For me as a reviewer it is difficult to complete a comprehensive review since the document I can download, includes 4 different colours, the new version is overwritten in the old version which difficults the correction.

Anyhow it looks that the authors have addressed the comments from the previous review. However, the editors should check the final format. For me, as far as I understand, this requires at least minor revision of the final version.

Reviewer 3 Report

I read the revised paper entitled “Dynamic Proteomic Analysis of White Hypsizygus marmoreus in the Whole Process of Mushroom Cultivation” submitted to Journal of Fungi. I regard that the problems on this manuscript have already been revised.